# Local Control Following Stereotactic Body Radiation Therapy for Liver Oligometastases: Lessons from a Quarter Century

Sara Mheid [1], Stefan Allen [2], Sylvia S. W. Ng [3], William A. Hall [4], Nina N. Sanford [5], Todd A. Aguilera [5], Ahmed M. Elamir [5], Rana Bahij [6], Martijn P. W. Intven [7], Ganesh Radhakrishna [8], Issa Mohamad [9], Jeremy De Leon [10], Hendrick Tan [11,12], Shirley Lewis [13], Cihan Gani [14], Teo Stanecu [1], Veronica Dell'Acqua [15] and Ali Hosni [1,*]

1    Department of Radiation Oncology, University of Toronto, Princess Margaret Cancer Centre, Toronto, ON M5G 2M9, Canada; sara.mheid@uhn.ca (S.M.); teodor.stanescu@uhn.ca (T.S.)
2    Department of Radiation Oncology, Dalhousie University, Nova Scotia Health, Halifax, NS B3H 4R2, Canada; stefanc.allen@nshealth.ca
3    Department of Radiation Oncology, University of Toronto, Odette Cancer Centre, Sunnybrook Health Sciences Centre, Toronto, ON M4N 3M5, Canada; sylvia.ng@sunnybrook.ca
4    Department of Radiation Oncology, Medical College of Wisconsin, Milwaukee, WI 53226, USA; whall@mcw.edu
5    Department of Radiation Oncology, University of Texas Southwestern, Dallas, TX 75235, USA; nina.sanford@utsouthwestern.edu (N.N.S.); todd.aguilera@utsouthwestern.edu (T.A.A.); ahmed.elamir@utsouthwestern.edu (A.M.E.)
6    Department of Oncology, Odense University Hospital, 5000 Odense, Denmark; rana.bahij@rsyd.dk
7    Department of Radiotherapy, Division Imaging and Oncology, University Medical Centre, 3584 CX Utrecht, The Netherlands; m.intven@umcutrecht.nl
8    Department of Radiotherapy, The Christie NHS Foundation Trust, Manchester M20 4BX, UK; ganesh.radhakrishna2@christie.nhs.uk
9    Department of Radiation Oncology, King Hussein Cancer Center, Amman 11941, Jordan; imohamad@khcc.jo
10   GenesisCare, Alexandria, NSW 2015, Australia; jeremy.deleon@genesiscare.com
11   Department of Radiation Oncology, Fiona Stanley Hospital, Perth, WA 6150, Australia; hendrick.tan@genesiscare.com
12   GenesisCare, Perth, WA 6150, Australia
13   Department of Radiotherapy and Oncology, Manipal Comprehensive Cancer Care Centre, Kasturba Medical College, Manipal Academy of Higher Education, Manipal 576104, India; shirley.lewis@gmail.com
14   Department of Radiation Oncology, University Hospital Tübingen, 72076 Tübingen, Germany; cihan.gani@med.uni-tuebingen.de
15   Medical Affairs and Clinical Research, Linac-Based RT, Elekta Milan, 20864 Lombardy, Italy; veronica.dellacqua@elekta.com
*    Correspondence: ali.hosni@rmp.uhn.on.ca; Tel.: +1-416-946-2124; Fax: +1-416-946-6561

**Abstract:** The utilization of stereotactic body radiation therapy for the treatment of liver metastasis has been widely studied and has demonstrated favorable local control outcomes. However, several predictive factors play a crucial role in the efficacy of stereotactic body radiation therapy, such as the number and size (volume) of metastatic liver lesions, the primary tumor site (histology), molecular biomarkers (e.g., KRAS and TP53 mutation), the use of systemic therapy prior to SBRT, the radiation dose, and the use of advanced technology and organ motion management during SBRT. These prognostic factors need to be considered when clinical trials are designed to evaluate the efficacy of SBRT for liver metastases.

**Keywords:** liver; SBRT; oligometastases

## 1. Introduction

The liver is a common site of metastasis for several malignancies, most commonly from the primary tumor of the colorectum, breast, lung, stomach, esophagus, pancreas, and melanoma origin [1–3]. The most common source of liver metastasis is colorectal cancer (CRC),

and up to 50% of these patients develop metastasis to the liver [3]. Additionally, an analysis of the Surveillance, Epidemiology, and End Results (SEER) database (2010–2015) showed that the overall incidence rate of liver metastases was 22.66 per 1,000,000 individuals [4].

In the past decades, local treatment of the liver oligometastasis has become increasingly common. The term "oligometastasis" was coined by Weichselbaum and Hellman in 1995 as a state between the absence of metastasis and the diffuse spread of disease [5]. In 2011, Weichselbaum and Hellman defined oligometastasis as metastasis limited in number and distribution as per standard diagnostic imaging scans [6,7]. In 2020, ESTRO-ASTRO published a consensus document defining oligometastatic disease as "1–5 metastatic lesions, a controlled primary tumor being optional, but where all metastatic sites must be safely treatable" [8].

Surgical resection, whenever feasible, is the mainstay treatment for medically operable patients with resectable liver oligometastases [9]. For those with liver metastases not suitable for surgical resection, localized treatments such as radiofrequency ablation (RFA), microwave ablation (MWA), or stereotactic body radiation therapy (SBRT) can be considered [1,10,11].

The Deutschen Gesellschaft für Radioonkologie (DEGRO) defined SBRT as a dedicated form of external beam radiation therapy with specific physical and biological characteristics. It is marked by a sharp transition in radiation dose outside the treated tumor while increasing the dose inside the specified tumor volume. This treatment modality is combined with image guidance techniques and advanced motion management to yield effective local treatment for liver oligometastases with a relatively low toxicity rate [11–13]. The biological principle behind SBRT is believed to trigger specific radiobiological pathways, enhancing both localized tumor control and systemic treatment outcomes [7]. The aim of this review is to provide an overview of the local control (LC) following SBRT for liver metastases.

## 2. Importance of LC of Liver Metastases (Mortality, Widespread Progression and Morbidity)

The LC for liver metastasis following SBRT is defined as a lack of progression of the metastatic liver lesion that was treated with SBRT (i.e., any response or stable disease). Achieving LC of liver oligometastases with SBRT has been associated with improved overall survival (OS) [11,14,15]. Klement et al. reported the outcomes of 388 patients with 500 metastatic lesions (lung: $n = 209$, liver: $n = 291$) treated with SBRT. Their findings suggested that achieving LC using SBRT improved OS in CRC patients with lung or liver metastases and a projected OS estimate of more than 12 months [16]. Kok et al. also reported that a higher dose of SBRT (BED10 > 100 Gy10) significantly improved LC of liver metastases (compared to lower dose SBRT $\leq$ 100 Gy10), which resulted in significant improvement of OS (85% vs. 48%, $p = 0.007$) [17]. Furthermore, the study conducted by McPartlin et al. demonstrated similar significant findings, indicating an association between improved OS and LC of targeted liver disease ($p = 0.001$) [11]. More recently, Cao et al. published their multi-institutional retrospective review of 1700 extracranial oligometastases in 1033 patients (25.2% non-small cell lung cancer, 22.7% CRC, 12.8% prostate cancer, and 8.1% breast cancer) [18]. Patients who experienced local failure within six months of SBRT for any oligometastasis had a 3.6-fold higher risk of mortality and 2.7-fold higher risk of wide-spread progression in comparison to those who maintained the local control of treated oligometastasis ($p < 0.001$) [18]. Regardless of the specific time point within the 3-year post-SBRT period that was analyzed, similar trends and relationships between LC duration and outcomes were consistently observed.

In an analysis of morbidity associated with locally uncontrolled liver metastases, several clinical and laboratory signs were reported, including deterioration of liver function test (79.2%), liver failure (10.4%), jaundice (28.6%), cachexia (17.3%), hepatic encephalopathy (9.1%), abdominal distension (8.7%), hepatomegaly (11.3%), pain (10.4%), infectious disease (5.6%), respiratory distress (7.8%), and gastrointestinal bleeding (1.3%) [19]. On the other hand, modern technology-based liver SBRT is well tolerated with minimal occurrence of treatment-related adverse effects. Acute toxicity, characterized by occasional transient

deterioration in liver function tests, was described in other studies [13,20,21]. Furthermore, several studies reported no or very rare incidence of radiation-induced liver disease (RILD) [2,20,22]. According to Andratschke et al., less than 1% of cases experienced grade 3 acute toxicity [12]. In a systematic review, Mutsaers et al. demonstrated well-preserved post-liver SBRT quality of life (QOL) [23]. Mendez Romero et al. prospectively assessed the impact of SBRT on the QOL of patients with primary and metastatic liver tumors and found no statistically significant decline in mean QOL over six months [24].

## 3. Comparison of LC of Liver Metastases Following SBRT and Other Ablative Therapies

It is important to note that there are limited randomized data comparing SBRT to other local modalities for the treatment of liver metastases. Several retrospective and prospective studies reported favorable LC rates following SBRT for limited liver metastases [2,15,20–22,25–39]. Table 1 summarizes key retrospective and prospective studies published in the last two decades. Petrelli et al. published a systematic review in 2018, including 18 studies of SBRT to liver metastasis, and reported one- and two-year LC rates of 67% and 59.3%, respectively [40].

**Table 1.** Outcomes of liver SBRT from selected retrospective and prospective series.

| Study | Type | Number of Lesions | Size (cm/mL) | Radiation Dose | BED10 | Follow Up (Median) | 1 y LC/OS | 2 y LC/OS |
|---|---|---|---|---|---|---|---|---|
| Rusthoven et al. [2] | Prospective | 49 | Median 14.9 mL (0.75–97.98) | 60 Gy in 3 fractions | 180 Gy10 | 16 months | 95% LC | 92% LC 30% OS |
| Lee et al. [25] | Prospective | 143 | Median 134.8 mL (6.7–3090) | 54–60 Gy in 6 fractions | 102.6–120 Gy10 | 10.8 months | 71% LC 63% OS | - |
| Scorsetti et al. [15] | Prospective | 76 | 1.1–5.4 cm/ (1.8–134.3) mL | 75 Gy in 3 fractions | 262.5 Gy10 | 6.1 years | 95% LC 85% OS | - |
| Herfarth et al. [32] | Prospective | 56 | ≤6 cm/ 10 (1 to 132) mL | 14–26 Gy in 1 fraction | 33.6–93.6 Gy10 | 5.7 months | 71% LC 72% OS | - |
| Kavanagh et al. [35] | Prospective | 21 patients | 14 (1–98) mL | 60 Gy in 3 fractions | 180 Gy10 | 19 months | 93% LC | - |
| Mendez Romero et al. [36] | Prospective | 34 | 3.2 (0.5–7.2) cm/ 22.2 (1.1–322) mL | 37.5 Gy in 3 fractions | 84.38 Gy10 | 12.9 months | 94% LC 85% OS | 82% LC 62% OS |
| Hong et al. [37] | Prospective | 143 | 2.5 cm (0.5–11.9 ) | 30–50 GyE (Proton SBRT) in 5 fractions | 100.72, 48 Gy10E | 30.1 months | 71.9% LC 66.3% OS | 35.9% OS |
| Folkert et al. [38] | Prospective | 39 | 2.0 cm (0.5–5.0 cm) | 35–40 Gy in 1 fraction | 157.5–200 Gy10 | 25.9 months | - | 96.6% 4 yLC 82% OS |
| Rule et al. [39] | Prospective | 37 | 2.5 cm (0.4–7.8) | 30–60 Gy in 5 fractions | 40–132 Gy10 | 20 months | 72% LC | 57.6% OS |
| Wulf et al. [20] | Retrospective | 23 | 50 mL (9–516) | 30 Gy in 3 fractions | 60 Gy10 | 9 months | 76% LC 71% OS | 61% LC 43% OS |
| Katz et al. [28] | Retrospective | 174 | 2.7 cm (0.6– 12.2) | 30–55 Gy in 5 fractions | 48–115.5 Gy10 | 14.5 months | 76% LC | 57% LC |
| Cazic et al. [22] | Retrospective | 16 patients | ≤6 cm | 60 Gy in 8 fractions | 105 Gy10 | 12 months | 62.5% LC 87.5% OS | - |
| Coffman et al. [29] | Retrospective | 81 | 2.5 cm (0.7–8.9) | 36–60 Gy in 3 fractions (proton SBRT) | 60–180 Gy10E | 15 months | 92.5% LC | - |

Jackson et al. compared the outcomes of patients with liver metastases from various malignancies treated with either RFA (*n* = 112) or SBRT (*n* = 170). In lesions measuring ≥ 2 cm, SBRT was associated with improved freedom from local progression, with LC rates reaching 96% and 88.2% at one and two years, respectively, whereas RFA exhibited lower rates of 74.7% and 60.6% at the respective time points [41]. Another retrospective study compared treatment outcomes of RFA (*n* = 268) vs. SBRT (*n* = 62) for CRC liver metastases and showed that for tumors > 2 cm, SBRT demonstrated significantly better LC compared to the RFA group (*p* < 0.001) [26]. In 2019, a meta-analysis and systematic review compared RFA vs. SBRT for liver malignancies and included three studies for liver metastases. LC rate at two years was higher in the SBRT group (*n* = 909) compared to the RFA group (*n* = 1329) (83.6% vs. 60%,

$p < 0.001$) [42]. A single institutional retrospective study by Franzese et al. compared the outcomes of patients with liver metastases from CRC treated with either MWA or SBRT. The one-year LC was higher in the SBRT group compared to the MWA group in tumors greater than 3 cm in size (91% vs. 84%, $p = 0.02$) [43].

### 4. Radiographic Evaluation of Local Control Following SBRT for Liver Metastases

The modified Response Evaluation Criteria in Solid Tumors (mRECIST) is an updated framework for assessing tumor response to treatment that incorporates a comprehensive evaluation of treatment response in patients with solid tumors [44]. Following high-dose radiation therapy to the liver, CT radiographic changes with distinct appearances can be observed [45,46]. Herfarth et al. categorized these changes into three types based on the timing after SBRT. Type I was observed during the initial 1–2 months, with a hypodense appearance on portal-venous CT images and an isodense appearance on delayed contrast CT images. Type II occurred approximately 3–4 months after SBRT, with the irradiated zone exhibiting hyperdensity on delayed images. Finally, type III was observed at approximately six months post-SBRT and beyond, with the more developed area becoming either isodense or hyperdense on the portal-venous phase and persistently hyperdense on the late contrast phase [45,46].

Functional MRI images such as diffusion-weighted imaging have been shown to be a valuable tool for differentiating between various pathological manifestations, including edema, necrosis, hemorrhage, recurrence, and cystic changes [47,48]. Solanki et al. and Stinauer et al. suggested that 18F-FDG PET-CT is a useful tool for evaluating the response to SBRT in liver metastases, particularly when CT and MR imaging features present difficulties in interpretation [49,50].

Following SBRT for liver metastases, some tumors exhibit minimal changes in enhancement and signal intensities and insignificant changes in size, especially when imaging is conducted within a short period (i.e., <3 months) post-treatment. Thus, more extensive follow-up imaging studies may be necessary to accurately evaluate the response of liver metastases to SBRT [47]. Aitken et al. recommended follow-up imaging at three-month intervals during the first year and at six-month intervals thereafter [51]. Similarly, Herfarth et al. conducted imaging follow-up five to ten weeks after SBRT, with subsequent scans scheduled at intervals of three to five months [46].

### 5. Factors Affecting LC Following SBRT for Liver Metastases

*5.1. Tumor-Related Factors*

5.1.1. Primary Tumor Type

Primary tumor origin and histology have been shown to impact outcomes amongst patients who received SBRT for liver metastases [12,52,53] (Table 2). The German group analyzed 474 patients with liver oligometastases from various histologies. Primary tumor origin was a significant predictor of LC. Metastases from CRC had a significantly worse LC rate at one year (67%) compared to breast cancer (91%), non-small cell lung cancer (88%), or other histologies (80%) [12]. Similarly, Ahmed et al. analyzed 372 liver metastases from various primary cancers following SBRT and reported significantly poorer LC rates for liver metastases of CRC origin, with one- and two-year LC rates of 79% and 59% for CRC lesions, compared to 100% for non-CRC lesions ($p = 0.02$) [54,55].

**Table 2.** Summarizes prospective and retrospective series on liver SBRT from various primary sites.

| Primary Site | Author | Design | No. of Patient | No. of Liver Lesions | Dose | Median Follow Up (mo) | Median OS (mo) | 1 y LC/OS | 2 y LC/OS | 1/2 y PFS |
|---|---|---|---|---|---|---|---|---|---|---|
| CRC | Hoyer et al. [33] | Phase 2 | 44 | 141 | 45 Gy/3 fx | 52 | 19.2 | 67% OS | 86% LC 38% OS | 19% 2 y |
| | Vanderpool et al. [31] | Prospective | 20 | 31 | 12.5–15 Gy/3 fx | 26 | 34 | 100% OS 100% LC | 74% LC 83% OS | - |
| | Chang et al. [27] | Retrospective | 65 | 102 | 22–60 Gy/1–6 fx | 14 | - | 72% OS 62% LC | 45% LC 38% OS | - |
| | Py et al. [21] | Retrospective | 67 | 99 | 37.5–54 Gy/ 3–5 fx | 47 | 53 | 95.5% OS 86.6% LC | 72.4% LC 81.4% OS | 81% 1 y 54% 2 y |
| | Voglhuber et al. [30] | Retrospective | 115 | 150 | 35 Gy/5 fr | 11.4 | 20.4 | 72% OS 82% LC | 82% LC 45% OS | 20% 1 y 10% 2 y |
| | Yu et al. [26] | Retrospective | 44 | 62 | 36–60 Gy/3–5 fr | 31.8 | - | 96% OS | - | - |
| Esophageal | Li et al. [56] | Retrospective | 8 (liver) | - | Median BED10 60 (39–90 Gy) | 35 | 14 | - | - | - |
| BTC | Franzese et al. [57] | Retrospective | 21 (liver) | - | Median 45 Gy (24–75)/3–10 fx | 14 | 13.7 | 58% OS 76.7% LC | 71% LC 41% OS | 36% 1 y 20%: 2 y |
| Pancreatic ADC | Lee et al. [58] | Retrospective | 76 | - | Median 50 Gy (40–50)/5 fx | 10.9 | 8.5 | 38% OS 66% LC | - | 7%: 1 y |
| GI NET | Hudson et al. [59] | Retrospective | 25 | 53 | Median 50 Gy/ 5 fr (25–60 Gy/3 fr) | 14 | - | 92% LC | - | 44%: 1 y |
| GYN | Laliscia et al. [60] | Retrospective | 8 (liver) | - | 24 Gy/1 fr, 27 Gy/3 fr | - | - | - | - | - |
| Breast | Milano et al. [61] | Prospective | 14 (liver) | 33 (curative) | - | - | - | - | 76% OS | 44%: 2 y |
| | Onal et al. [62] | Retrospective | 22 | 29 | 54 Gy/3 fx | 16 | - | 85% OS 100% LC | 88% LC 57% OS | 38%: 1 y 8%: 2 y |
| | Franzese et al. [63] | Retrospective | 54 | - | Median dose 60 Gy (30–75 Gy)/ 3 fx (3–6 fx) | 26.2 | - | 95.5% OS 57.6% LC | 41.6% LC 76.9% OS | 38.7%:1 y 22%: 2 y |
| | Tan [64] | Retrospective | 120 (24 liver) | 29 | 30–60 Gy/ 3–5 fx | 15.25 | 53.16 | 83.5% OS 89% LC | 86.6% LC 70% OS | 45%: 1 y 32%: 2 y |
| RCC & Melanoma | Stinauer et al. [65] | Retrospective | Melanoma 17 RCC 13 | 11 | 40–50 Gy/5 fr, 42–60 Gy/3 fx | 28 | 24.3 | 88% LC | - | - |
| | Grossman et al. [66] | Retrospective | 16 RCC 15 melanoma | 14 (liver) | Median SBRT 50 Gy, median fractional dose 5 Gy | - | 10.8 | 94.7% LC | - | - |
| Melanoma | Franceschini et al. [67] | Retrospective | 31 (8 liver) | 11 | 50.25–75/3–6 fx | 13 | 10.6 | 41%OS 96.6% LC | 82.8% LC 21% OS | 18.5%: 1 y 13.9%: 2 y |

Abbreviations: CRC: colorectal cancer, BTC: biliary tract cancer, ADC: adenocarcinoma, GI NET: gastrointestinal neuroendocrine tumor, GYN: gynecological malignancies, RCC: renal cell carcinoma.

The impact of primary tumor type on LC of liver metastases after SBRT extends beyond the origin of the tumor and histologic subtype to the molecular phenotype of the cancer. Hong et al. analyzed the association of genetic alterations with LC in 89 patients (CRC being the most common primary) treated with proton-based liver-directed SBRT. They reported lower LC for lesions with KRAS mutation (one-year LC of 43% compared to 72%, $p = 0.02$) and reduced LC rates in cases with both KRAS and TP53 mutations (one-year LC of 20% compared to 69%, $p = 0.001$) [37,68].

5.1.2. Number, Size, and Volume of Metastatic Liver Lesions

Joo et al. reported that the number of treated liver lesions (one, two, or three) was a significant predictive factor for intrahepatic tumor control. Their findings suggest that an increasing number of treated sites was associated with a higher risk of reduced intrahepatic control [69]. Several trials had set a maximum tumor diameter of <6 cm for high-dose liver SBRT [2,15,25,70,71]. Doi et al. retrospectively reviewed the records of 24 patients with 39 metastatic liver tumors from CRC who were treated with SBRT. On multivariable analysis, a maximum tumor diameter $\leq 3$ cm was significantly associated with better LC ($p = 0.03$) [72]. Similarly, Rusthoven et al. reported that for lesions with a maximum diameter $\leq 3$ cm, 2-year LC was 100% compared to 77% for lesions > 3 cm ($p = 0.015$) [2].

Andratschke et al. reported that smaller treated metastatic tumor volume, as observed in the study where the GTV volume ranged from a minimum of 0.6 cc to a maximum of 699 cc (median volume of 27 cc), and the PTV volume ranged from a minimum of 4.5 cc to a maximum of 1074.0 cc (median volume of 71.3 cc), were found to be a significant predictor for LC ($p < 0.001$) [12]. Furthermore, Flamarique et al. reported that tumor volumes > 30 cc correlated with worsened two-year LC rates (90% vs. 34.5%) ($p = 0.005$) [73].

### 5.2. Treatment-Related Factors

#### 5.2.1. Prior Liver-Directed Local Therapies

There is evidence to suggest that SBRT is a safe and well-tolerated treatment option with excellent LC rates for liver metastases that have been previously treated with other liver-directed therapy, including surgery, ablation, and transarterial chemoembolization (TACE) [74,75]. Moon et al. conducted a prospective single-arm trial to evaluate the safety and efficacy of liver SBRT in patients with or without prior liver-directed therapy. The study included a total of 30 patients, among whom 63% had liver metastases (47% had received prior liver-directed therapies, which included liver resection, TACE, and RFA). Out of the 30 patients, 28 underwent SBRT to a new lesion, and 2 received SBRT due to either a local recurrence or a sub-optimal response following TACE. The study did not find a statistically significant difference in LC between those who had previously undergone liver-directed therapies and those who had not (73% and 86% at one year, respectively, $p = 0.70$) [75].

#### 5.2.2. Pre-SBRT Systemic Therapy

The surviving tumor cells, after systemic therapy, may develop a better ability to repair DNA damage, which results in a more radioresistant phenotype [52,76]. This may explain why metastatic cancer that was previously treated with adjuvant systemic therapy tends to be more aggressive [52,77]. Klement et al. analyzed 623 liver metastases in 464 patients who underwent SBRT treatment from any histology-proven primary solid tumor [52]. They found that patients who had received chemotherapy before SBRT had significantly reduced LC at two years compared to those who did not (58% vs. 83%, $p = 0.04$) [52]. Sheikh et al. conducted a multi-institutional retrospective analysis, which included 235 patients with a total of 381 CRC oligometastatic lesions treated with SBRT. On multivariable analysis, they found that receiving any systemic therapy before SBRT was linked to an increased risk of progression ($p < 0.001$) [55]. Furthermore, Andratschke et al. reported that patients who received systemic therapy before SBRT had worse LC rates compared to those who did not [12]. It is important to consider multiple factors when interpreting this correlation, as patients who received systemic therapy first might have had a higher burden of the disease. Future research employing novel biomarkers of disease burden (e.g., ctDNA analysis) is warranted.

#### 5.2.3. SBRT Dose

Multiple prospective phase I/II trials for liver metastases have shown two-year LC rates ranging from 60% to 100% with different radiation dose and fractionation schedules. A phase II study by Scorsetti et al. treated 61 patients with liver metastases from different primary histologies with a dose of 75 Gy in three fractions showed a three-year LC of 78%, with no significant difference in LC based on histology (CRC vs. other) and size of lesion (>3 cm vs. <3 cm) when ultra-high dose SBRT was used [71]. For lesions measuring <3 cm, an ablative dose of 60 Gy delivered over three fractions yielded LC rates of 95% and 92% at one and two years, respectively [2]. For tumors measuring <6 cm, a higher ablative dose of 75 Gy given over three fractions achieved an LC rate of 94% [15]. A phase I/II dose escalation SBRT study by Rusthoven et al. showed two-year LC rates of 92% [2]. McPartlin et al. showed lower LC rates of 50% and 26% at one year and four years, respectively, in CRC liver metastases, likely due to lower SBRT dose used (median, minimum SBRT dose was 37.6 Gy (range, 22.7–62.1 Gy) in 6 fractions) [11].

Retrospective and modeling studies have shown improved LC with a high BED10 dose for liver metastases [78]. A single-institution retrospective study by Kok et al. showed a two-year LC of 90% with BED10 > 100 Gy10 vs. 60% with BED10 < 100 Gy10 [17]. Similar results were shown by Mahadevan et al. with BED10 > 100 Gy10 (wot-year LC 77.2% vs. 59.6%) in 427 patients with liver metastases [79]. Ohri et al. described better LC outcomes with BED10 > 100 Gy10, and the tumor control probability (TCP) modeling showed two-year LC increased to 76% at BED10 of 100 Gy10 and 90% with BED10 of 180 Gy10 [80]. While higher radiation doses could lead to better LC, it should be recognized that there is a tendency for high radiation doses to be prescribed for small tumors.

While the literature on single fraction SBRT is limited and early studies included both hepatocellular carcinoma (HCC) and liver metastases, more recent studies of single fraction SBRT for liver metastases with dose escalation to 35–40 Gy demonstrated two-year LC of 100% and four-year LC of 96.6% with no reported grade 3 toxicity [38,81]. Folkert et al. used a 35–40 Gy single fraction and applied the following constraints: max point dose 14 Gy, 12.4 Gy, 15.4 Gy, and 18.4 Gy to the spinal cord, stomach, and duodenum, jejunum, and colon, respectively. Furthermore, 700 mL of uninvolved liver received <9.1 Gy [38]. The results from these studies suggest that higher BED10 delivered in a single fraction can provide excellent LC with acceptable toxicities. Prospective randomized phase III trials are needed to further evaluate the efficacy and toxicity of single fraction ultra-high dose SBRT.

Despite the accumulating evidence that showed the association between SBRT dose and LC of liver metastases, it is not always possible in clinical practice to treat all liver metastases with very high dose SBRT mainly due to the proximity of the tumor to the central biliary tree or luminal structures (e.g., small/large bowel, stomach and duodenum). When selecting high-dose SBRT, it is also important to consider the volume of the uninvolved liver (i.e., whole liver minus gross tumor volume [GTV]) and the pre-SBRT liver function to avoid potential liver toxicity [2,11,12,73,82].

### 5.2.4. Advanced Organ Motion Management

Organ motion management is critical in liver SBRT in order to safely deliver ablative doses to the target while limiting the volume of normal tissue irradiated [83,84]. During liver SBRT, it has been reported that the internal motion of tumors can reach up to 39.5 mm (mean 17.6 mm) [85]. Various motion management mechanisms can be used, such as fiducial markers, abdominal compression, breath hold techniques, gating, and tumor tracking [83,86,87]. Imaging studies for motion measurement and evaluation include fluoroscopy, 4D CT, 4D cone-beam computer tomography scans, 2D cine MR, and 4D MR imaging [1,88–90]. Several studies have demonstrated that the use of advanced motion management techniques in liver SBRT is associated with improved LC of liver metastases [12,91]. In 2014, a report was published highlighting the importance of adequate respiratory motion management in SBRT for oligometastatic CRC patients. Results showed that metastases in moving organs (e.g., liver) exhibited an LC of 53% at 1 year compared to 79% for lymph nodes ($p = 0.01$) [92]. Klement et al. also reported that simple motion management techniques (such as free breathing and abdominal compression) predicted significantly lower tumor control probability [52].

### 6. Future Directions

The use of SBRT in the management of liver oligometastases has shown promising results. However, there are still many areas that require further research and development in order to optimize its use and effectiveness. One future direction in the management of liver oligometastases using SBRT includes the use of combination therapies, such as SBRT in combination with immunotherapy or targeted therapies, to improve the outcomes. Data from various types of cancer suggest that the combination of immunotherapy and SBRT is well-tolerated in metastatic disease [10,93–95].

The development of advanced imaging and planning technologies can improve the accuracy and precision of SBRT delivery. This includes the use of real-time imaging and

tracking, as well as the integration of machine learning algorithms to improve treatment planning and response evaluation. MR-guided radiotherapy is a cutting-edge and rapidly developing technology that allows for improved visualization of tumors and surrounding normal tissue during treatment, resulting in highly precise treatment delivery, even with moving targets. Online adaptive SBRT planning may reduce the risk of radiation-related toxicities and increase the dose delivered to the tumor [96]. Rosenberg et al. found that 75% of metastatic liver lesions (44% of which were CRC metastases) showed no local progression following MR-guided liver SBRT (median dose 50 Gy in five fractions) at a median follow-up of 21 months [97]. Given the promising outcomes observed in various studies [89,97–102], the MAESTRO phase II prospective trial is designed to evaluate the potential benefits of adaptive MR-guided SBRT compared to conventional SBRT administered at a standard linear accelerator for patients with liver metastases [103].

Proton beam therapy is another radiation therapy modality that allows more liver sparing given the rapid dose falloff beyond the edge of the target [104]. Hong et al. studied 89 patients with liver metastases from various types of cancer were enrolled. Of these patients, 38.2% had primary CRC. The median tumor size was 2.5 cm, and the radiation dose administered ranged from 30 to 50 GyE, delivered in five sessions. The one-year and three-year LC rates for the entire group were 71.9% and 61.2%, respectively [37]. Furthermore, Colbert et al. published their experience treating five patients with right hemi-liver proton therapy for bilobar colorectal liver metastases who were ineligible for second-stage hepatectomy; 67.5 cobalt gray equivalents in 15 daily fractions of proton therapy, using a deep inspiration breath-hold technique, was given to the right hemiliver with concurrent capecitabine. Local control (radiographic partial or complete response) was evident in all patients except for one who was treated with a BED of 89.6 Gy. In all patients, radiation therapy was well tolerated without substantial toxicity [105].

Similarly, carbon ion radiotherapy (C-ion RT) offers the ability to achieve precise dose localization through the Bragg peak [106]. Shiba et al. reviewed 102 patients with oligometastatic liver disease who had received C-ion RT between May 2016 and June 2020. The median dose was 60 Gy (58–76 Gy), and the median tumor size was 27 mm (7–90 mm). Local control rates at one and two years were 90.5% and 78%, respectively, and none of the patients experienced grade 3 or higher acute or late toxicities [106].

Another evolving area of research involves the use of selective radiation dose boosting to improve tumor control, particularly in combination with functional imaging biomarkers (e.g., fluorodeoxyglucose, fluoromisonidazole, or other PET radiotracers). This approach could help overcome radioresistance associated with tumor metabolic activity or hypoxia [68]. Building on this concept, Popple et al. studied the effect on tumor control probability (TCP) of increasing the dose to hypoxic areas and reported that a boost dose (ranging from 120% to 150% of the initial dose) increased TCP [107].

## 7. Conclusions

The utilization of SBRT for the treatment of liver oligometastases has demonstrated favorable LC outcomes. However, several factors influence the efficacy of SBRT, such as the number and size (volume) of liver lesions, the primary tumor type (origin/histology/phenotype), the use of systemic therapy prior to SBRT, the radiation dose, the use of advanced technology and organ motion management during SBRT. These factors need to be considered when clinical trials are designed to evaluate the efficacy of SBRT for liver metastases.

**Author Contributions:** Conceptualization, S.M., A.H., S.A., S.S.W.N., W.A.H., N.N.S., T.A.A., R.B., M.P.W.I., G.R., I.M., J.D.L., H.T., S.L., T.S., C.G. and V.D.; writing—original draft, S.M., A.H., S.A., S.S.W.N., W.A.H., N.N.S., T.A.A., R.B., M.P.W.I., G.R., I.M., J.D.L., H.T., S.L., T.S., C.G., V.D. and A.M.E.; writing—review and editing, S.M., A.H., S.A., S.S.W.N., W.A.H., N.N.S., T.A.A., R.B., M.P.W.I., G.R., I.M., J.D.L., H.T., S.L., T.S., V.D., C.G. and A.M.E. All authors have read and agreed to the published version of the manuscript.

**Funding:** This research received no external funding.

**Data Availability Statement:** This study did not report any data.

**Conflicts of Interest:** Ali Hosni: non-financial leadership, DSC of liver TSG at ELEKTA MRL consortium.

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
