# Peer review of "Local Control Following Stereotactic Body Radiation Therapy for Liver Oligometastases: Lessons from a Quarter Century"

_curroncol, doi:10.3390/curroncol30100667_

Round 1

Reviewer 1 Report

This is a well-written and thorough review on the topic of SBRT for liver oligomets, and should be valuable for anyone seeking guidance on treatment options. My only comment is that the two tables, which are extremely important, are difficult to read and perhaps could be better organized. At a minimum, both should be printed in landscape. For Table 1, since the last two columns (LC and OS) both generally have 1 year and 2 year components, it might make more sense to have those two columns be split as 1 year LC/OS and 2 year LC/OS instead. Then you can remove all the "1 year" and "2 year" text from inside the table. You could also remove text by abbreviation "fractions" as "fx." For Table 2, there are a lot of acronyms that may be obvious but are not defined (BTC, ADC, NET, RCC). Table 2 also needs to be printed landscape and hopefully without the weird lines I see through the text. Maybe there's also room here to combine or simplify some of the columns like LC/OS/PFS to make it more readable.

Author Response

Thank you very much for kindly reviewing the manuscript. Your input is highly appreciated and your comments regarding the table reformatting have been taken into consideration and have been added to the updated manuscript.

Reviewer 2 Report

This review article is detailed and informative about the utility of SBRT. It would be good for young radiation oncologists to read it for educational purposes.

A chapter of 1-2 paragraphs explaining the biological and physical principles of SBRT would be added. There are only text and tables, so the readability is a bit low. It would be nice if you could add a case picture or graphic related to SBRT.

It is true that an increase in radiation dose leads to a higher LC. However, it should be recognized that there is a tendency for high radiation doses to be prescribed for small tumors. Please add this too.

Author Response

Thank you so much for kindly reviewing the manuscript. Your input is greatly appreciated. I have added the information regarding the biological and physical principles of SBRT and I have also added your valuable comment about increased radiation dose and higher local control.

Reviewer 3 Report

The authors provide a comprehensive survey of what is known about stereotactic radiotherapy for oligo liver metastases. The paper is very informative for clinicians, as it compiles information necessary for actual treatment, including comparisons between stereotactic radiotherapy and other treatment modalities, post-treatment evaluation methods, and prescribed doses.

Page 9, lines 254 to 260

For safe treatment, a more specific description of dose constraints for the gastrointestinal tract and normal liver would be helpful.

Page 10, lines 300 to 313

It is good that the proton therapy is mentioned, but it would be better if the carbon-ion therapy is also mentioned.

Please check this paper (Shiba S, Wakatsuki M, Toyama S, et al. Carbon-ion radiotherapy for oligometastatic liver disease: A national multicentric study by the Japan Carbon-Ion Radiation Oncology Study Group (J-CROS). Cancer Sci. 2023;00:1-8. doi:10.1111/cas.15) and others.

Author Response

Thank you very much for your valued input. I really appreciate the time you took to review the manuscript. I have included more specific details about dose constraints and I have also included the reference you have kindly added regarding carbon ion therapy.

Reviewer 4 Report

I see an obvious problem with this manuscript: the issue concerns radioembolization (TARE) in hepatocellular carcinoma, while the manuscript concerns stereotactic radiotherapy in liver metastases. Both the therapy and the clinical setting do not match.

Author Response

Thank you so much for kindly reviewing the manuscript. 

The academic editor replied last time and confirmed that the manuscript will be continues processing as regular article in Current oncology “not for HCC issue”